# High-Resolution Magic-Angle Spinning Nuclear Magnetic Resonance Identifies Impairment of Metabolism by T-2 Toxin, in Relation to Toxicity, in Zebrafish Embryo Model

**DOI:** 10.3390/toxins16100424

**Published:** 2024-10-01

**Authors:** Ariel Lawson, Mark Annunziato, Narmin Bashirova, Muhamed N. Hashem Eeza, Jörg Matysik, A. Alia, John. P. Berry

**Affiliations:** 1Department of Chemistry and Biochemistry, Florida International University, Miami, FL 33181, USA; alaws016@fiu.edu (A.L.); mannu002@fiu.edu (M.A.); 2Institute of Environment, Florida International University, Miami, FL 33181, USA; 3Institute for Analytical Chemistry, University of Leipzig, 04103 Leipzig, Germany; narmin.bashirova@medizin.uni-leipzig.de (N.B.); m.noureeza@gmail.com (M.N.H.E.); joerg.matysik@uni-leipzig.de (J.M.); 4Institute for Medical Physics and Biophysics, University of Leipzig, 04107 Leipzig, Germany; alia.aliamatysik@medizin.uni-leipzig.de; 5Leiden Institute of Chemistry, Leiden University, 2333 Leiden, The Netherlands; 6Biomolecular Science Institute, Florida International University, Miami, FL 33199, USA

**Keywords:** nuclear magnetic resonance (NMR), metabolomics, zebrafish, trichothecene, T-2 toxin

## Abstract

Among the widespread trichothecene mycotoxins, T-2 toxin is considered the most toxic congener. In the present study, we utilized high-resolution magic-angle spinning nuclear magnetic resonance (HRMAS NMR), coupled to the zebrafish (*Danio rerio*) embryo model, as a toxicometabolomics approach to elucidate the cellular, molecular and biochemical pathways associated with T-2 toxicity. Aligned with previous studies in the zebrafish embryo model, exposure to T-2 toxin was lethal in the high parts-per-billion (ppb) range, with a median lethal concentration (LC_50_) of 105 ppb. Exposure to the toxins was, furthermore, associated with system-specific alterations in the production of reactive oxygen species (ROS), including decreased ROS production in the liver and increased ROS in the brain region, in the exposed embryos. Moreover, metabolic profiling based on HRMAS NMR revealed the modulation of numerous, interrelated metabolites, specifically including those associated with (1) phase I and II detoxification, and antioxidant pathways; (2) disruption of the phosphocholine lipids of cell membranes; (3) mitochondrial energy metabolism, including apparent disruption of the tricarboxylic acid (TCA) cycle, and the electron transport chain of oxidative phosphorylation, as well as “upstream” effects on carbohydrate, i.e., glucose metabolism; and (4) several compensatory catabolic pathways. Taken together, these observations enabled development of an integrated, system-level model of T-2 toxicity in relation to human and animal health.

## 1. Introduction

Species of the fungal genus *Fusarium*, as well as several related taxa within the order Hypocreales, are producers of a family of mycotoxins known as *trichothecenes*, which are known to contaminate a wide range of agricultural crops and products. Of these, T-2 toxin is generally considered the most toxic [1], and it is recognized to contaminate grains and cereals, including animal feeds [2,3], which may contaminate livestock (i.e., meat and dairy products), and in the case of aquaculture, fish and other seafood.

Exposure to T-2 toxin can cause a range of adverse effects in both humans and animals. The reported effects include both acute symptoms, ranging from vomiting, diarrhea and abdominal discomfort to skin irritation, disorientation, headache and fever, as well as death, and chronic long-term impacts including wasting (i.e., weight loss), immunosuppression and reproductive dysfunction [4,5]. With respect to animal health, this toxicity can consequently reduce productivity in both livestock and aquaculture [2,3]. At a cellular level, T-2 toxin causes oxidative stress, disrupts cell membrane integrity, and prevents protein synthesis [4]. The peptidyl transferase activity of the ribosome is inhibited by the T-2 toxin’s binding to the 60S subunit of the ribosome, which prevents polypeptide chain elongation [6,7]. Reactive oxygen species (ROS), produced in response to exposure to T-2, on the other hand, can harm cellular macromolecules, including lipids, proteins and DNA [8]. Despite the number of recognized mechanisms of action of T-2, however, a comprehensive system-level model of toxicity with respect to the adverse effects has remained elusive.

Metabolomics, which enables identification and quantitation of differences in a wide range of “small molecule” metabolites—corresponding to an equally diverse range of metabolic pathways—has emerged as an important tool in the study of toxins, so-called toxicometabolomics, with respect to identifying the biochemical, molecular and cellular targets and, moreover, pathways of toxicity. Within this field, the use of nuclear magnetic resonance (NMR), and in particular, high-resolution magic-angle spinning (HRMAS) NMR, has emerged as a powerful tool for characterizing the pathways and developing integrated, system-level models of toxicity [9]. In order to reduce, or largely remove, interactions, including anisotropic, dipolar, and quadrupolar effects, HRMAS NMR utilizes specially designed field gradients and sample spinning at the so-called “magic angle” (m = 54.74°) [10] to enable well-resolved NMR spectra of chemical mixtures within complex matrices, and in particular, solid-state samples, including biological tissues [11]. In doing so, it enables highly reproducible quantitation of major metabolites spanning a diversity of pathways within complex, biological matrices, including in vivo assessments of intact samples (e.g., cells, whole organisms).

High-resolution MAS NMR has recently been developed and applied to investigate how a wide array of environmental toxicants affect, in particular, the metabolic profiles in the early life (i.e., embryonic and larval) stages of zebrafish [9,12,13,14,15,16,17,18]. Owing to a number of practical advantages, including the size, transparency, and rapid development, the zebrafish embryo has been widely established as a vertebrate model for toxicology, and particularly, environmental toxicology [19,20], including applications in metabolomics and other “omics”, e.g., genomics, transcriptomics, proteomics, and as a potential model for human health effects [21,22,23,24]. Notably, previous studies have, in fact, utilized this approach for understanding the toxicity of many important mycotoxins [25,26,27] and the underlying metabolic alterations associated with these fungal-derived toxicants [14,17,28,29]. As not only an established animal model but also, and perhaps more importantly, a relevant ecological receptor for these toxicants, studies in the zebrafish and other teleost fish species have enabled both characterization of integrated, system-level models of toxicity, and simultaneously, identification of potential biomarkers that are, in turn, relevant for assessing both exposure and effect.

In the present study, we utilized HRMAS NMR, coupled to the zebrafish embryo model (Figure 1), for metabolic profiling to investigate the toxicity and, moreover, develop an integrated system-level model of toxicity for T-2 toxin.

## 2. Results

### 2.1. Toxicity of T-2 in Zebrafish Embryos

The dose-dependent embryotoxicity, as specifically measured by lethality (over 24 h), of T-2 was observed (Figure 2) over a range of 5 to 500 parts-per-billion (ppb) for 72 h post-fertilization (hpf) zebrafish embryos with no observed adverse effect limits (NOAEL) and lowest observed adverse effect limits (LOAEL) of 40 and 50 ppb, respectively. A corresponding median lethal concentration (LC_50_) value of 105 ppb was calculated.

### 2.2. HRMAS NMR Metabolic Profile of T-2 Treated Embryos

Analysis by HRMAS NMR resolved (Figure 3) and reproducibly quantified 40 metabolites (Table 1) for 72 hpf embryos exposed to 50 ppb T-2 toxin for 24 h. Principal components analysis (Figure 4A), and subsequent statistical analyses (Figure 4B), indicated significant differences for 18 of these metabolites in embryos exposed to 50 ppb T-2 (for 24 h) versus the control (i.e., solvent vehicle only) embryos, spanning a range of cellular and biochemical pathways. Of these, significant decreases were observed for four metabolites related to carbohydrate metabolism, including glucose (Glc), glucose-1-phosphate (G1P) and glucose-6-phosphate (G6P), as well as lactate (Lac), while an increase was observed for pyruvate (Pyr). At the same time, two metabolites involved in the tricarboxylic acid (TCA), i.e., “Kreb’s”, cycle were significantly altered in the T-2 exposed embryos: significantly lower levels were also observed for malate (Mal), while succinate (Succ) significantly increased. While no significant difference in the ATP or ADP levels were observed, both creatine (Cr) and phosphocreatine (pCr) increased. The levels of acetate (Ace), which is associated with a range of cellular functions, including energy metabolism, in particular, increased in the T-2 treated embryos. The levels of three amino acids were significantly altered by T-2: alanine (Ala), lysine (Lys) and phenylalanine (Phe) decreased, while aspartate (Asp) increased. Aligned with the possible detoxification and/or oxidative stress, the levels of the reduced form of glutathione (GSH) increased. Related perhaps to the effects on the cell membrane integrity, metabolites associated with phospholipid membranes, including glycero-3-phosphate (GPC) and O-phosphocholine (PChol), both decreased. Notably, the liver-specific metabolite trimethylamine N-oxide (TMAO) increased, as did the neurotransmitter γ-aminobutyrate (GABA), compared to the untreated (solvent-only) controls.

### 2.3. Production of ROS in T-2-Treated Zebrafish

The altered production of ROS was observed, using a fluorescence assay, in the 72 hpf zebrafish exposed to 200 ppb T-2 compared to the untreated controls (Figure 5). While both the control and T-2-treated embryos showed evidence of ROS production in the gastrointestinal (GI) tract, the treated embryos showed decreased—and essentially undetectable—production in the upper GI, specifically the intestinal bulb, compared to the untreated controls. Similarly, whereas the ROS production was pronounced in the developing liver of the control embryos, this was not observed in the T-2-treated embryos. In contrast, however, a substantial increase in ROS production was observed in the brain region of the treated embryos, whereas no such apparent production was observed in the control embryos.

## 3. Discussion

The embryonic stages of zebrafish have been shown to be an exceptional model for understanding the toxicity and the associated adverse effects of a broad range of mycotoxins as contaminants of agricultural products in relation to human and animal, e.g., livestock and aquaculture, health [27]. While relatively few studies have employed the zebrafish embryo model to investigate the trichothecene mycotoxins, including T-2, at least two previous studies have documented embryotoxicity, including developmental toxicity and behavioral abnormalities, in zebrafish embryos exposed to T-2 and other trichothecenes (e.g., type B deoxynivalenol [DON]). Yuan et al. [30] reported a range of developmental abnormalities and accompanying behavioral changes in zebrafish embryos (over 1 to 6 dpf), and related these effects to cellular apoptosis and oxidative stress (i.e., ROS production). More recently, Khezri et al. [26] similarly reported the developmental toxicity of T-2, as well as DON, including delayed hatching rates in the zebrafish embryo model, but they were not able to observe previously reported behavioral effects. The results of the present study contribute to our understanding of the toxicity of T-2, particularly in the zebrafish embryo model, and specifically provide insight, through application of HRMAS NMR-based metabolomics, into the underlying cellular, molecular and biochemical processes associated with toxicity and other adverse effects.

### 3.1. Embryotoxicity and ROS Production in T-2-Exposed Zebrafish Embryos

Aligned with previous studies [26,30], the dose-dependent toxicity (i.e., lethality) for T-2-exposed zebrafish embryos was measured in the 50 to 300 ppb range (Figure 2). A calculated LC_50_ value of 105 ppb was generally comparable to, albeit slightly lower than, the values measured at equivalent stages (i.e., 72–96 hpf embryos) in previous studies (e.g., ~144 ppb, [30]; ~191 ppb, [26]). Based on this, an exposure concentration of 50 ppb, corresponding to the LOAEL for embryotoxicity, was selected for the subsequent HRMAS NMR studies (discussed below).

Alongside toxicity, oxidative stress was assessed in the T-2-exposed, compared to the solvent-only control, embryos at the 72–96 hpf stage, specifically based on fluorescent detection of ROS production (Figure 5). Notably, while the ROS production within the GI tract was observed for both the treated and control embryos, substantial fluorescence (i.e., ROS) was additionally observed in the developing liver of the untreated controls, but this was not detectable in the T-2-exposed embryos. High rates of production of ROS in the liver are generally expected due to high rates of energy production by mitochondria in hepatocytes, which is the major source of ROS [31]. The lack of detectable ROS production in the T-2-treated embryos, on the other hand, suggests a possible shift in energy metabolism, and perhaps impaired mitochondrial energy production, which is, likewise, suggested by a number of associated metabolites (as discussed below). In contrast, the T-2-treated embryos showed a substantial increase in ROS production in the brain region, while production was essentially undetectable in the brain of the untreated controls. This observation aligns with the documented ability of T-2 to effectively cross the blood–brain barrier (BBB) and the reported neurotoxicity that has, in fact, been linked to oxidative stress and damage, as well as mitochondrial dysfunction [8,32]. The differential effects on the liver and brain, including the role of oxidative stress and mitochondrial, and the corresponding energy metabolism impairment is, in turn, borne out by many of the changes in the HRMAS NMR metabolic profiles observed in the T-2-exposed zebrafish embryos and supports the development of an overall system-level model of T-2 toxicity (Figure 4 and Figure 5; as discussed further below).

### 3.2. Integrated Model of T-2 Toxicity Based on HRMAS NMR Metabolic Profiling

Metabolomics, and other “omics” approaches, enable a system-level approach to understanding the interconnectivity of cellular, molecular and biochemical pathways toward achieving a holistic picture of biological processes, including toxicity and the associated adverse health effects. Coupling HRMAS NMR, as well as other analytical platforms (e.g., mass spectrometry), to the early life stages of the zebrafish has previously proven effective as a means of developing integrated models of toxicity in relation to a broad range of environmental toxicants [9,12,13,14,15,16,17,18]. Within the context of embryotoxicity and the apparent role of system-specific ROS production associated with the exposure of zebrafish embryos to T-2, the observed alteration in metabolites measured in the present study identified several potential targets and system-specific pathways of toxicity, including a role of the xenobiotic detoxification pathways and mitochondrial dysfunction in relation, in particular, to energy metabolism and consequent oxidative stress. Toward an integrated model of T-2 toxicity (Figure 6 and Figure 7), the role of each of these targets and pathways is discussed below.

#### 3.2.1. Detoxification and Antioxidant Pathways

Exposure to T-2 was accompanied by notable changes in metabolites linked to the upregulation of the phase I and II detoxification pathways, which primarily occur in the liver, as well as associated antioxidants, which are variably found in different cell types. Firstly, a significant increase in TMAO is generally suggestive of increased phase I detoxification in hepatocytes. Derived from trimethylamine that is produced, in turn, from several metabolic and, particularly, dietary precursors (e.g., choline, carnitine, betaine) by the gut microbiota, TMAO is known to be a product of flavin-containing monooxygenases (FMOs), which—similar to cytochrome P450 (CYP) enzymes—are ascribed to the biotransformation of xenobiotics, particularly in the liver, as part of phase I detoxification. Similarly localized to hepatocytes, TMAO has been, accordingly, identified [14] as a metabolic biomarker of liver function. The observed increase in TMAO, therefore, is specifically suggestive of an upregulation in the phase I detoxification pathways in response to T-2. Previous studies [1,33] have, in fact, identified a number of presumptive phase I metabolites in both in vitro (i.e., hepatocytes) and in vivo studies, including, most notably perhaps, the toxic deacylated congeners, HT-2 and neosolaniol [34], as well as a number of other hydroxylation products. Although increased FMO has not been specifically shown in response to the T-2 toxin, such a response would align with the reported upregulation of the more well-studied phase I monooxygenase, CYP1A1 [35]. Interestingly, TMAO is not only a biomarker of hepatic function, but elevated levels of this metabolite have, in turn, been linked to the dysregulation of energy metabolism, including lipid and glucose regulation [36], and may additionally explain several of the altered metabolites (as discussed below) in this regard. Moreover, alongside these phase I products, a number of presumptive phase II metabolites of T-2, including, in particular, glucuronides, were reported [33], as has been the upregulation of several associated conjugation enzymes (e.g., glucuronosyltransferase; [37]) in association with T-2 exposure.

The observed increase in GSH (Table 1), likewise, suggests either upregulated phase II detoxification, or alternatively, a response to oxidative stress. Indeed, a number of studies of T-2 have similarly reported increases in GSH, alongside changes in a number of associated detoxification and antioxidant pathways, particularly in various fish and avian systems [38,39,40,41]. And it has been similarly shown [30] that exogenous GSH supplementation of zebrafish embryos exposed to T-2 attenuates developmental toxicity associated with ROS production. It is proposed that the increased GSH observed here corresponds to the upregulation of phase I and II detoxification in hepatocytes in response to exposure without attending glutathione conjugation or antioxidant activity. The lack of reduced glutathione consumption, particularly as part of antioxidant activity, aligns with the apparently reduced ROS production in the liver of the T-2-exposed embryos (Figure 5), which is perhaps due to dysfunction in mitochondrial energy production suggested, as discussed below (see Section 3.2.3. Energy Metabolism), by the alteration of a number of metabolites. Moreover, studies of T-2 toxin metabolism have, in fact, not identified any GSH conjugates, but rather, glucuronides exclusively [33].

In contrast to the liver, the brain region of the T-2-exposed embryos showed a pronounced increase in ROS (Figure 5). While inhibition of energy metabolism, and particularly, the electron transport chain (ETC) of oxidative phosphorylation, as proposed for the T-2 toxin (discussed below), may be expected to depress mitochondrial ROS production in certain cell types, such as the liver—as a primary site of catabolism—inhibition of complex II of ETC (as an established target of T-2) has been shown to be highly nuanced with respect to its role in the production of ROS [42,43,44]. This observed difference in ROS production may not be surprising, therefore, particularly given that neural cells are characterized by the highest energy demands compared to other cell types (e.g., hepatocytes) and specifically rely on the energetic pathways associated with high oxygen consumption [45,46,47]. It is, furthermore, known that T-2 readily crosses the BBB [48], and moreover, whereas hepatocytes are the main site for metabolic detoxification and subsequent elimination of T-2 [1], the toxin has been shown to accumulate in the brain [32]. Perhaps most revealing in this context is the observed significant increase in GABA (Figure 4 and Table 1). A preponderance of evidence suggests that oxidative stress, and specifically, ROS, significantly alters GABAergic pathways, including increased release and binding and subsequently decreased uptake, all of which are accompanied by increased levels of the neurotransmitter [49]. At the same time, a number of studies have reported the neurotoxicity of T-2, with a role of oxidative stress specifically linked to these effects (see the recent review by [29]). Increased GABA is, furthermore, notable given that no significant changes in NAA, an established biomarker of neural cells, were observed: this observation suggests that while increased ROS and, perhaps, the consequent effects on certain (e.g., GABAergic) pathways occur, there is no apparent neural cytotoxicity (i.e., cell death).

#### 3.2.2. Cell Membrane Integrity and Phospholipid Pathways

Significant increases in GPC and PChol are indicative of a role of membrane disruption, and perhaps related phospholipid metabolism, in the toxicity of T-2. Metabolic precursors and, alternatively, degradation products of phosphatidylcholines (PC)—as the primary glycerophospholipids of eukaryotic cells—GPC and PChol are recognized as potential biomarkers of cell membrane integrity [50]. Decreased levels of PC were, in fact, very recently observed in avian hepatocytes exposed to T-2 and suggested to result from peroxidation of membrane lipids in association with increased ROS production [51]. On the other hand, PChol and GPC are recognized to be key components of the turnover of PC via the CDP-choline pathway (or so-called “Kennedy Pathway”) that includes not only synthesis of PC but also breakdown to GPC and PChol, which have, in turn, been shown to be important for maintaining homeostasis in proliferation cells [52]. Notably, GPC and PChol are direct products of the phospholipase neuropathy target esterase (NTE), which has a critical role in the embryonic development of the nervous system [53,54,55]. It has been suggested that oxidative stress may, in turn, be linked to NTE-based neuropathy, particularly as a recognized target of organophosphates [56], aligned with the increased production of ROS in the embryonic brain of exposed zebrafish embryos, as observed here. At the same time, metabolites in the CDP-choline pathway have been shown to attenuate oxidative stress, and GPC has been reported, in particular, to stabilize complex I of the electron transport chain and reduce oxidative stress [57,58]. The latter observation is noteworthy given the apparent inhibition of succinate dehydrogenase (i.e., complex II) by the T-2 toxin, as identified in the present study (as discussed below) and in several other systems [5,59,60,61,62].

#### 3.2.3. Energy Metabolism

Most conspicuous, and perhaps central to the observed effects of T-2 on the zebrafish embryo, is the alteration of numerous metabolites associated with energy metabolism, and in particular, several mitochondrial pathways (Figure 7), underscoring the likely role of the mitochondria as a target of T-2, as likewise suggested by several previous studies [1,8,63,64,65,66,67].

Paramount, perhaps, to the observed impacts of T-2 on energy metabolism is the apparent inhibition of succinate dehydrogenase (SDH). Inhibition of SDH is specifically evidenced in the present study by the increased concentrations of the substrate, Succ, suggesting an impediment at this key step (Figure 7), which is involved in both the TCA cycle, and concurrently, the ETC of oxidative phosphorylation (i.e., complex II). The specific inhibition of SDH is further suggested by a lack of observed significant change in other measured components of the TCA cycle preceding this step, including Cit and αKG (Table 1), as well as a concomitant decrease in Mal as a subsequent intermediate in the TCA cycle. And, indeed, it has long been recognized that T-2 inhibits SDH in relation to both the TCA cycle and as complex II of electron transport [5,59,60,61,62]. Consistent with the proposed impairment of oxidative phosphorylation, the presumptive inhibition of SDH would be notable as complex I/II is recognized to be the main contributor to the production of superoxide radicals during metabolism [68], and impairment of this component of the electron transport chain is, in fact, associated with both increased and decreased ROS [44,69,70]. Impairment of energy production in the brain and liver of developing embryos via SDH/complex II, therefore, may be reflected in the observed increased and decreased production of ROS in these regions, respectively (Figure 5; as discussed above).

The inhibition of SDH is, in turn, linked to a number of other changes in the metabolic profile of T-2-exposed embryos with respect to energy metabolism. Among the relevant changes, decreases in Glc, as well as G6P and G1P, are particularly noteworthy. These changes perhaps suggest increased utilization of glucose via both glycolysis (as indicated by decreased G6P), and perhaps glycogen breakdown (as indicated by decreased G1P, which is exclusively associated with glycogenolysis and glycogenesis), as a compensatory mechanism to offset the dysfunction in energy production via the TCA cycle and oxidative phosphorylation. This would be particularly true for cells, such as neural cells, with the highest glucose demand (to maintain a high energy demand) and perhaps less so for cells such as hepatocytes, which utilize a wider array of catabolic precursors. At the same time, an increase in both the Pyr and acetate signal (likely indicative of acetyl CoA) is consistent with increased glycolysis (to generate ATP), alongside decreased shunting of these glycolytic products to the TCA cycle. Tangentially, the concomitant decrease in Lac observed is suggestive of likely increased lactate dehydrogenase (LDH) activity, further increasing the Pyr levels and generating additional extramitochondrial NADH.

Alternatively, or in addition to their utilization in energy production, the depletion of Glc, G1P and G6P might represent the increased flux of these precursors via UDP-glucose toward glucuronidation (Figure 7). This might be particularly true for hepatocytes, given that glucuronides of T-2 have been shown to be the primary phase II conjugation products observed [33]. Indeed, previous studies have shown that several drugs known to be substrates for glucuronidation (and which deplete UDP-glucuronic acid by as much as 98%) can deplete both UDP-glucose and glycogen in hepatocytes by as much as 50%, suggesting a direct link between glucuronidation activity and glucose homeostasis [71]. The depletion of glucose substrates for energy metabolism (via redirection to detoxification pathways), and in particular, oxidative phosphorylation, would also be consistent with the observed decrease in ROS production observed in the liver of the T-2-exposed embryos: ETC in the mitochondria of hepatocytes is the primary source of ROS production, and whereas dysfunction in complex II due to binding of T-2 would be expected to lead to increased ROS (as observed in the brain), upstream substrate (i.e., glucose) depletion might explain—alongside utilization of alternative catabolic pathways—the reduced ROS production in hepatocytes (from aberrant oxidative phosphorylation).

Notably, while the decreased entry of Pyr and acetyl CoA into the TCA would be expected to lead to the impairment of energy production in the mitochondria, no significant changes in either ATP or ADP, nor the total NADH, were observed, suggesting a possible role of alternative, compensatory energy-producing (i.e., ATP) pathways. This is particularly true given the dual role of SDH in the TCA cycle and, concurrently, as complex II of oxidative phosphorylation. Possible alternative pathways, in this regard, include substrate-level phosphorylation to generate ATP either via increased glycolysis or mitochondrial pathways, including succinyl-CoA ligase, which can be used to phosphorylate ADP, particularly when the mitochondria are in an otherwise “low-energy state” and the inorganic phosphate is high [72]. The simultaneous lack of significantly altered NADH levels, despite the presumptive impairment of the TCA cycle, would specifically align with increased glycolysis, which would generate both ATP and NADH. Further aligned with increased production of NADH via glycolysis (and other cytosolic pathways, e.g., LDH), a concomitant decrease in Mal, and an increase in Asp, suggests an enhanced role of the malate/aspartate shuttle as a means of transporting glycolysis-derived NADH from the cytosol to the mitochondrial matrix. An increased influx of cytosolic NADH could, in turn, either be utilized to compensate for reduced oxidative phosphorylation capacity, or alternatively, serve to bypass blockage of the TCA cycle at SDH (as well as the subsequently blocked entry of Pyr into the TCA cycle), and drive ATP production via substrate-level phosphorylation via succinyl-CoA ligase (Figure 7).

At the same time, a significant increase in the total creatine (*p* < 0.005), including both pCr and Cr, was observed, whereas no significant change in Crn (the breakdown product of pCr) was observed: an increased pool of total creatine could be indicative of an expanded use of transphosphorylation by creatine phosphokinase to generate ATP from ADP, which is recognized as an important pathway for tissues that rapidly consume ATP, including neurons. Aligned with the observed increase in the total creatine (tCr), creatine is biosynthesized from the amino acid, Gly, and *when normalized by tCr*, Gly is, indeed, significantly decreased (*p* < 0.05) in T-2-exposed embryos. Taken together, these observations suggest a possible, additional compensatory mechanism to offset impaired ATP production (via oxidative phosphorylation).

Finally, a significant decrease in amino acids, including Lys and Phe, is, likewise, suggestive of a shift in energy metabolism. The decreased levels of these specifically suggest a likely increased catabolism of these amino acids as essential amino acids. As amino acids are primarily catabolized in the liver, these decreases might specifically represent increased shunting toward energy metabolism in light of the otherwise impaired energy metabolism (i.e., glucose depletion, impaired TCA cycle and oxidative phosphorylation) in hepatocytes, as previously discussed. While it remains unclear exactly how the catabolism of these amino acids might support dysfunction in energy metabolism, it is notable that (1) Phe catabolism leads to production of Fum, which could offset reduction of this key TCA intermediate, owing to the presumptive inhibition of SDH; and (2) Lys catabolism, which occurs within the mitochondrial matrix, generates two equivalents of NADH, as well as acetate, and may serve to compensate for the impairment of oxidative phosphorylation. The possible catabolism of amino acids in the developing zebrafish embryo is particularly salient as it is known that amino acids (via various catabolic pathways) can account for up to 85% of energy requirements in teleost fish compared to, for example, mammals, for which amino acid catabolism accounts for only about 20% of energy metabolism [73].

## 4. Conclusions

High-resolution magic-angle spinning NMR has emerged as a powerful technique for metabolomics, particularly when coupled to cellular or in vivo, or otherwise intact, systems for elucidating integrated pathways associated with the toxicity of environmental contaminants, including, in the present study, mycotoxins as widespread contaminants of agricultural products in relation to human and animal health. The coupling of HRMAS NMR to the zebrafish embryo model, alongside assessments of the toxicity and relevant biochemical parameters (e.g., ROS production), enabled the proposal of an integrated, system-level model of T-2 toxicity. Central, perhaps, to this model is the suggested inhibition of SDH, as previously shown for the T-2 toxin [5,59,60,61,62] and evidenced here by the altered levels of both Succ and Fum, as well as Phe (as a metabolic shunt to circumvent this inhibition). Inhibition of SDH would lead to dysfunction, in turn, in both the TCA cycle and ETC of oxidative phosphorylation, impairing energy metabolism, including a shift to compensatory pathways (e.g., increased glycolysis, amino acid catabolism, upregulation of the Mal/Asp shuttle, increased utilization of transphosphorylation via creatine) and altered production of ROS, with possible implications for cellular integrity (e.g., membranes, oxidative stress) and system-specific processes (e.g., altered neurotransmission). In parallel, the apparent alteration of several indicators of increased phase I and II detoxification pathways, particularly in the liver, was observed, potentially overlapping with the altered energy metabolism pathways. With respect to the latter point, altered glucose metabolism, including Glc, G1P and G6P, is consistent with increased glucuronidation, which has previously been suggested to be a key pathway for the elimination of T-2. These findings not only improve understanding of the integrated pathways associated with T-2 toxicity but potentially provide insight for the future development of biomarkers of exposure and effect toward improved monitoring and mitigation in relation to both human and animal health with respect to this widespread mycotoxin.

## 5. Materials and Methods

### 5.1. Chemicals

A standard of the T-2 toxin was purchased from Cayman Chemical (Ann Arbor, MI, USA), and stock solutions (1 mg mL^−1^) were prepared in methanol for subsequent dilution into exposure media. All other chemicals were purchased from Thermo Fisher Scientific (Waltham, MA, USA), unless otherwise indicated.

### 5.2. Zebrafish Embryos

Zebrafish embryos (OBI/WIK line) for the HRMAS NMR studies were provided by the Helmholtz Centre for Environmental Research (UFZ; Leipzig, Germany). Rearing and breeding (to provide embryos) of zebrafish was performed as previously described [18]. Unfertilized, or dead or moribund, embryos were identified using a dissecting microscope and removed before transferring 120 embryos (for subsequent toxin exposure) to Petri dishes containing 40 mL of embryo media [74], which were placed in an incubator with a light cycle of 14 h light–10 h dark at 28 °C. All handling of the zebrafish, including rearing/breeding and exposure, and subsequent analyses, was carried out in accordance with German animal welfare guidelines and approved by the Government of Saxony, Landesdirektion, Leipzig, Germany (Aktenzeichen 75-9185.64).

To assess the T-2 toxicity, prior to the NMR studies, embryos of zebrafish (*Danio rerio*) from the PSA line were acquired from the University of Miami Rosenstiel School of Marine and Atmospheric Science (UM RSMAS). The zebrafish were reared and bred using established methods [75]. The collected eggs were washed with system water and placed in Petri dishes with about 30 mL of E3 medium [76], with approximately 120 eggs per plate. Prior to the 72 hpf exposures, the embryos were kept in an incubator (14 h light–10 h dark, 28 °C) and examined daily to remove any dead or moribund embryos. Unhatched embryos were transferred to 24-well test plates at 48 hpf (24 h prior to exposure), and subsequently, exposed to T-2 at 72 hpf (see 5.3. Assessment of Embryotoxicity, below) by dilution of stock solution into medium. Rearing and breeding was conducted based on protocols approved by the University of Miami’s Institutional Animal Care and Use Committee (UM IACUC, 20-006 LF), and they were performed by trained personnel.

### 5.3. Assessment of Embryotoxicity

To assess the toxicity and establish the exposure concentrations for the subsequent HRMAS-NMR experiments, the lethality of the zebrafish embryos (PSA line) was evaluated across a range of T-2 concentrations (5, 10, 20, 40, 50, 100, 200, 300 and 500 parts-per-billion [ppb]), as established in preliminary assessments and based on previous studies of T-2 toxicity in the zebrafish embryo [26,30]. The developmental stage (72 hpf) was chosen as it has previously been shown to enable sufficient development of key organ systems (e.g., CNS, liver/kidney, gastrointestinal) relevant to toxicity [16,17]; in addition, embryos are hatched at 72 hpf, and thus, exposure at this developmental stage eliminates any consideration of the chorion as a potential barrier to uptake. The exposures and toxicity assessments utilized previously established protocols [18] and were performed in triplicate (n = 3) using polypropylene 24-well plates (Evergreen Scientific, Los Angeles, CA, USA), with each well containing 5 embryos in 1.5 mL of E3 medium. The embryo lethality, including the corresponding no observable adverse effect level (NOAEL) and lowest-observable-adverse-effect level (LOAEL), was assessed over 24 h (until 96 hpf) by observing the cessation of movement, and absence/presence of a heartbeat, using a dissecting light microscope. The median lethal concentrations (LC_50_) were calculated using Probit analysis in SPSS (version 26.0; IBM Corporation, Armonk, NY, USA) and used, in turn, to guide the selection of exposure concentrations (and, specifically, sublethal concentrations) for the subsequent HRMAS NMR studies.

### 5.4. Exposure of Zebrafish Embryos, and Sample Preparation, for HRMAS NMR

For the HRMAS NMR metabolic profiling, zebrafish embryos (OBI/WIK line) were exposed to T-2 in accordance with previously established and validated protocols [18]. For each exposure, 120 embryos (72 hpf) were exposed to a nominal concentration of 50 ppb T-2 (alongside untreated, i.e., methanol-only, controls), based on the measured LC_50_ values (see 5.3. Assessment of Embryotoxicity), as a sublethal concentration. Any dead, or clearly moribund, embryos were removed during the course of the exposure to eliminate any metabolic affects associated with mortality. The embryos were exposed (at 72 hpf to 96 hpf) for 24 h in replicate (n = 6), alongside solvent-only controls (n = 6), in 25 mL of ISO medium in 100 mm polystyrene Petri dishes. The developmental stage (72–96 hpf) chosen has been previously shown to enable sufficient development of key organ systems (e.g., CNS, liver/kidney, gastrointestinal) relevant to toxicity [16,17], as well as to eliminate any barrier to uptake presented by the chorion (i.e., all the embryos are fully hatched by 72 hpf). At the end of the 24 h exposure period, 100 embryos were collected and washed with MilliQ water to remove excess T-2 and medium, and the collected embryos were subsequently “snap frozen” (to −80 °C) until analysis.

Prior to the NMR analysis, the embryos were transferred to a 4 mm zirconium oxide rotor (Bruker BioSpin AG, Fällanden, Switzerland), to which 10 μL of deuterated phosphate buffer (100 mM, pH 7.0) containing 0.1% (*w*/*v*) 3-trimethylsilyl-2,2,3,3-tetradeuteropropionic acid (TSP), as a chemical shift reference, was added. All the exposures (and handling) of the embryos were performed at the University of Leipzig and conducted in accordance with the German animal protection standards approved by the Government of Saxony, Landesdirektion, Leipzig, Germany (Aktenzeichen 75-9185.64) and the guidelines of the European Union, Directive 2010/63/EU, which explicitly permits the use of embryonic and early (≤5 dpf) larval stages of fish.

### 5.5. HRMAS NMR Analysis and Data Evaluation

HRMAS NMR was performed as previously described [18]. All the NMR investigations were performed using a Bruker DMX 600-MHz NMR magnet equipped with a 4 mm HRMAS dual ^1^H/^13^C inverse probe fitted with a magic-angle gradient and rotating at a rate of 6000 Hz. At a temperature of 277 K, measurements were performed using a Bruker BVT3000 control unit. The data acquisition and processing were carried out using the Bruker TOPSPIN program (Bruker BioSpin GmbH, Ettlingen, Germany). The 1H HR-MAS NMR spectra were acquired using the standard Bruker pulse sequence “zgpr” (from the Bruker pulse program library) for water suppression using pre-saturation pulses. Each one-dimensional spectrum was obtained using a spectral width of 12 kHz, 4 k time-domain (TD) data points, 128 averages, an acquisition time of 170 ms, and a relaxation delay of 2 s. Prior to the Fourier transformation, all the spectra were treated using an exponential window function corresponding to a 1 Hz line widening and zero-filled using TOPSPIN 3 (Bruker BioSpin GmbH, Germany). The total analysis time (including sample preparation, NMR parameter tuning, and data capture) for each ^1^H-HRMAS NMR spectroscopy copy was around 20 min. Sample spinning took 4 min 42 s, and there was no discernible effect on the embryo integrity.

Chenomx NMR Suite 9.0 was used to reference, baseline- and phase-correct, and quantify the metabolites (Chenomx Inc., Edmonton, AB, Canada). This was facilitated by matching each spectrum to a spectral signature from the Human Metabolome Database (HMDB). Chenomx’s 600 MHz library was used, which determines the concentration of individual molecules based on the concentration of a known reference signal (in this case, TSP). The Profiler module was used to profile the HRMAS NMR spectra from embryos. To quantify each component in a spectrum, a list of unique chemical shifts is required, along with the peak shape, compound description, and basic signal information. The presence of a compound is proven when all the signal areas have integrals proportional to their chemical shifts and peak shape. Subsequently, the concentrations of metabolites were determined using the total sum of metabolites for normalization. OriginPro v10 and Metaboanalyst (https://www.metaboanalyst.ca/) was used to perform statistical analysis on the NMR quantification.

### 5.6. Visualization of Reactive Oxygen Species (ROS) in Zebrafish Embryos

To supplement the HRMAS NMR studies, particularly in relation to the proposed oxidative stress, the production of reactive oxygen species (ROS) was observed and quantified in zebrafish embryos using intracellularly oxidized 2′,7′-dichlorofluorescein (DCF) as a fluorescent probe, as described previously [18]. For these analyses, embryos were exposed to 200 ppm T-2 toxin for 24 h, alongside untreated (i.e., solvent only) controls. At 96 hpf, the embryos were treated with 1 mM (in 4% DMSO) of the nonfluorescent cell-permeable probe chloromethyl-2′,7′-dichlorofuorescein diacetate (CM-H_2_DCFA; Invitrogen^TM^, Carlsbad, CA, USA) at a maximum concentration of 10 M, and then incubated for 60 min. After 60 min, the embryos were washed three times to remove CM-H_2_DCFA from the medium and transferred to a glass slide with a borosilicate coverslip. The anesthetic ethyl 3-aminobenzoate methanesulfonate (MS-222, 1 mg/mL) was added to each sample to immobilize the embryos, as described previously [18]. Images were taken with an inverted laser-scanning confocal microscope (Leica DMi8/TL LED, Leica Microsystems CMS GmbH), with an excitation wavelength of 485 nm and an emission wavelength of 530 nm, equipped with a Leica HC PL Apo CS2 (5×/0.15 Dry) objective. The images were assessed using the Leica Application Suite X (LAS X) software package, version 3.1.5.

## Figures and Tables

**Figure 1 toxins-16-00424-f001:**
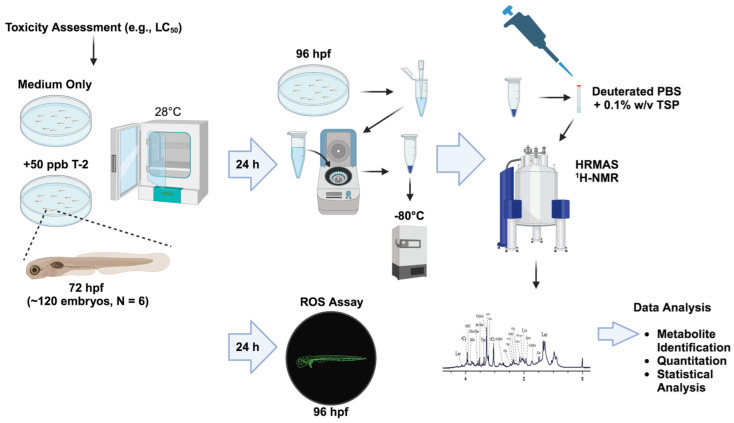
Workflow of the toxicometabolomics of the T-2 toxin in the zebrafish embryo model.

**Figure 2 toxins-16-00424-f002:**
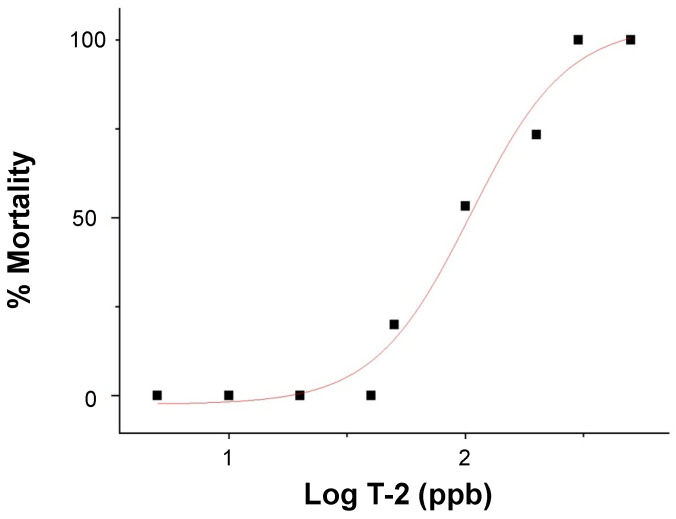
Dose-response curve of lethality (i.e., % mortality), in relation to the log exposure concentration of T-2 (in ppb), in 72–96 hpf zebrafish embryos.

**Figure 3 toxins-16-00424-f003:**
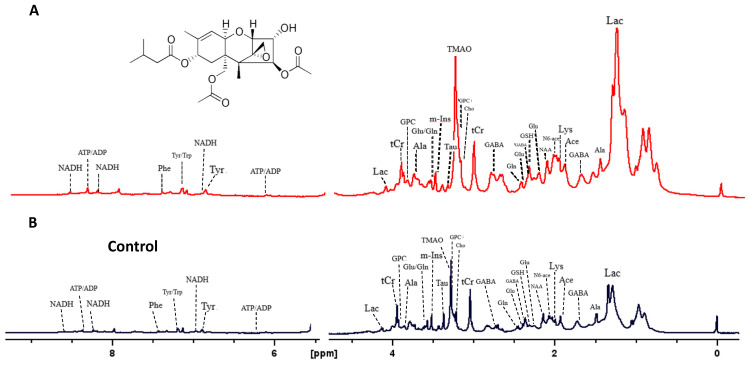
High-resolution MAS ^1^H-NMR spectra of zebrafish embryos exposed to the T-2 toxin (**A**), and control embryos exposed to ISO medium only (**B**).

**Figure 4 toxins-16-00424-f004:**
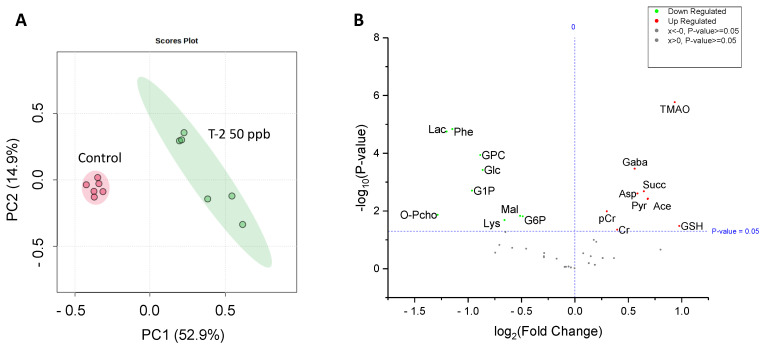
Principal components analysis (PCA) showing a significant difference in the metabolite concentrations of 50 ppb T-2-treated (green) versus untreated, i.e., solvent-only, control (pink) embryos of zebrafish (**A**), and volcano plot showing the log fold-change compared to the control embryos, and corresponding −log *p*-values, for metabolites identified and measured by HRMAS NMR. (**B**) As indicated, a significant change is reported for a *p*-value < 0.05. For a complete list of the metabolites measured, see Table 1. Abbreviations: Ace = acetate; Asp = aspartate; Cr = creatine; pCr = phosphocreatine; GABA = g-aminobutyric acid; Glc = glucose; GSH = glutathione; GPC = glycerophosphocholine; G1P = glucose-1-phosphate; G6P = glucose-6-phosphate; Lac = lactate; Lys = lysine; Mal = malate; O-PCho = O-phosphocholine; Phe = phenylalanine; Pyr = pyruvate; Succ = succinate; TMAO = trimethylamine oxide.

**Figure 5 toxins-16-00424-f005:**
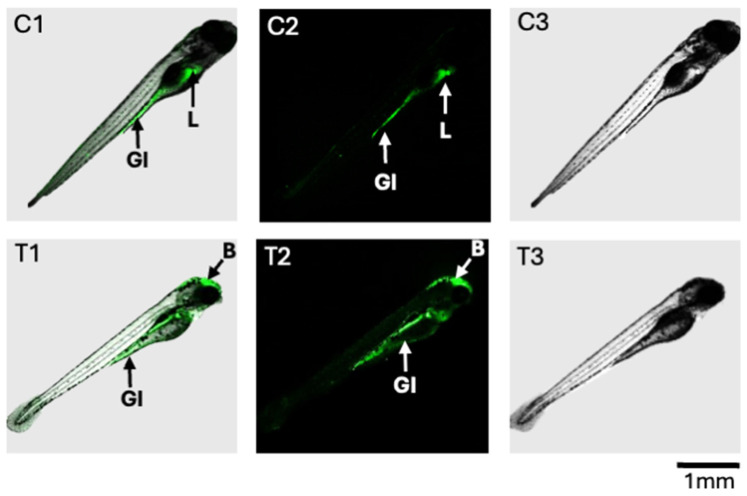
Fluorescence detection of ROS in zebrafish embryos at 96 hpf after 24 h exposure to T-2 (200 ppb). Shown are composite images (C1 and T1, respectively) as well as confocal fluorescence (C2 and T2, respectively) and light photomicrographs (C3 and T3, respectively) for the control and T-2-treated embryos. For the control embryos (C1 and C2), ROS were detected in the gastrointestinal (GI) tract and liver (L). For the T-2-treated embryos, ROS were, likewise, detected in the GI tract, but not the liver; however, ROS production was observed in the brain (B) region.

**Figure 6 toxins-16-00424-f006:**
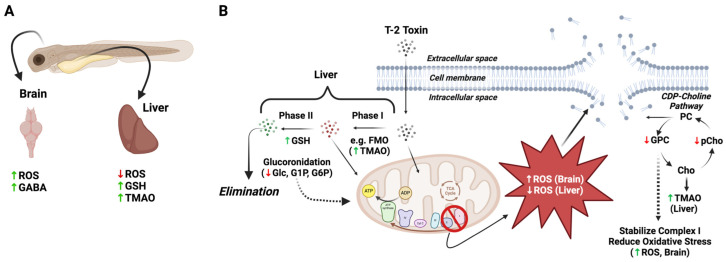
Proposed differential role of oxidative stress, detoxification pathways and disruption of cell membranes in T-2 toxicity in the brain and liver of zebrafish embryos. (**A**) Increased ROS was observed in the brain of T-2-exposed embryos, with an attending increase in γ-aminobutyric acid (GABA), presumably reflecting increased release and binding, and decreased uptake, whereas ROS was undetectable in the liver of T-2-exposed embryos, likely due to decreased mitochondrial energy production via oxidative phosphorylation. At the same time, the increase in both trimethylamine oxide (TMAO) and glutathione (GSH) presumably reflects increased phase I (i.e., flavin-containing monooxygenases [FMO]) and phase II detoxification pathways in the liver, respectively. (**B**) Increased phase II detoxification by glucuronidation, as previously reported [33], leads to elimination of T-2 in the liver, with a concomitant reduction in the pool of glucose (Glc) and related metabolites, i.e., glucose-1-phosphate (G1P) and glucose-6-phosphate (G6P), for mitochondrial energy production. Whereas ROS were decreased in liver, it is proposed that the increased ROS in the brain leads to the disruption of cell membranes, as suggested by the decreased levels of glycerophosphocholine (GPC) and O-phosphocholine (pChol), as part of the CDP-choline pathway for phosphatidylcholines (PC). It is recognized that GPC can stabilize complex I of the electron transport chain of oxidative phosphorylation, and reduced GPC levels may, accordingly, further explain the increased ROS in the brain region via “leakage” of electrons to O_2_. In the liver, increased production of TMAO may also deplete the levels of choline (Cho), as a precursor, which would further reduce the pools of GPC and pChol. Abbreviations for the metabolites: Ace = acetate; Asp = aspartate; Cr = creatine; pCr = phosphocreatine; GABA = g-aminobutyric acid; Glc = glucose; GSH = glutathione; GPC = glycerophosphocholine; G1P = glucose-1-phosphate; G6P = glucose-6-phosphate; Lac = lactate; Lys = lysine; Mal = malate; O-PCho = O-phosphocholine; Phe = phenylalanine; Pyr = pyruvate; Succ = succinate; TMAO = trimethylamine oxide.

**Figure 7 toxins-16-00424-f007:**
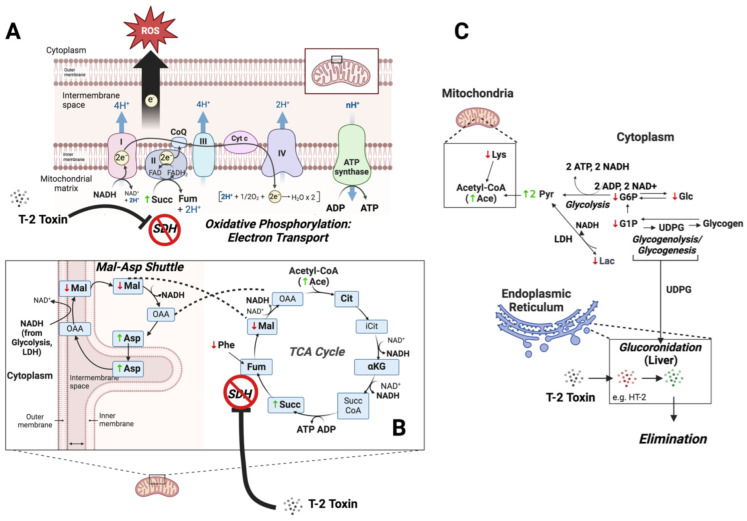
Proposed role of impaired energy metabolism in T-2 toxicity. Known to inhibit succinate dehydrogenase (SDH), the T-2 toxin impairs both complex II of the electron transport chain of oxidative phosphorylation (**A**), and a key step in the tricarboxylic acid (TCA) cycle (**B**), as suggested by the increase in succinate (Succ). Dysfunction in the electron transport chain is, in turn, associated with the increases and decreases in ROS observed in the brain and liver of exposed zebrafish, respectively (see Figure 5). A concomitant increase in the malate/aspartate shuttle, as reflected by decreased malate (Mal) and increased aspartate (Asp), compensates for impaired mitochondrial energy production by increasing the transport of cytoplasmic NADH from glycolysis and other pathways, e.g., lactate dehydrogenase (LDH), into mitochondria. Upstream of the impairment of these energy-generating pathways, decreased levels of glucose (Glc) and the associated metabolites, glucose-1-phosphate (G1P) and glucose-6-phosphate (G6P), reflect a compensatory increase in their catabolism to generate ATP via substrate-level phosphorylation (e.g., glycolysis), as well as possible shunting toward glucuronidation via UDP-glucose (UDGP) for phase II elimination of T-2 and possible phase I metabolites, e.g., HT-2, in the endoplasmic reticulum (**C**). Concurrent increases in pyruvate (Pyr) and acetate (as Acetyl CoA) reflect, in turn, reduced flux into mitochondrial pathways (i.e., TCA cycle) due to inhibition of SDH. Finally, decreased levels of essential amino acids, phenylalanine (Phe) and lysine (Lys), reflect increased catabolism toward compensatory generation of fumarate (Fum), as part of the TCA cycle, and acetyl CoA for subsequent entry into the TCA cycle, respectively.

**Table 1 toxins-16-00424-t001:** Metabolites altered by T-2, compared to the untreated (i.e., solvent-only) controls, based on HRMAS NMR measurement of zebrafish embryos. Treated embryos exposed to 50 ppb T-2. Given are the percent changes (% change) relative to controls and the corresponding *p*-values.

Metabolite (*Function*/*Role*)	% Change	*p*-Value
** *Energy Metabolism* **		
ADP	−5%	0.87
ATP	14%	0.74
NADH	−40%	0.28
**Glucose (Glc)**	**−45%**	**<0.001**
**Glucose-1-Phosphate (G1P)**	**−49%**	**0.002**
**Glucose-6-Phosphate (G6P)**	**−37%**	**0.02**
**Pyruvate (Pyr)**	**61%**	**0.003**
**Lactate (Lac)**	**−55%**	**<0.001**
**Acetate (Ace)**	**60%**	**0.004**
**Creatine (Cr)**	**32%**	**0.045**
**Creatine Phosphate (pCr)**	**23%**	**0.01**
Creatinine (Crn)	−18%	0.37
** *Energy Metabolism: TCA Cycle* **		
a-Ketoglutarate (aKG)	15%	0.12
Citrate (Cit)	75%	0.22
**Malate (Mal)**	**−29%**	**0.02**
**Succinate (Succ)**	**57%**	**0.002**
Fumarate (Fum)	−18%	0.29
** *Amino Acids* **		
Alanine (Ala)	0%	0.98
Asparagine (Asn)	29%	0.44
**Aspartate (Asp)**	**50%**	**0.002**
Glutamate (Glu)	11%	0.38
Glutamine (Gln)	13%	0.10
Glycine (Gly)	6%	0.30
**Lysine (Lys)**	**−30%**	**0.02**
**Phenylalanine (Phe)**	**−56%**	**<0.001**
Tryptophan (Trp)	−33%	0.19
Tyrosine (Tyr)	−11%	0.46
Taurine (Tau)	−18%	0.40
** *Cell Membranes* **		
Choline (Cho)	10%	0.65
**Glycerophosphorylcholine (GPC)**	**−46%**	**<0.001**
**O-Phosphocholine (pCho)**	**−59%**	**0.01**
** *Lipid Metabolism* **		
Carnitine	−2%	0.92
** *Oxidative Stress and Detoxification* **		
**Glutathione (GSH)**	**97%**	**0.03**
Carnosine	−39%	0.15
Anserine	−27%	0.21
** *Neural Cells* **		
N-Acetylaspartate (NAA)	20%	0.43
**g-Aminobutyrate (GABA)**	**48%**	**<0.001**
*myo*-Inositol	−36%	0.055
** *Hepatocytes* **		
**Trimethylamine N-Oxide (TMAO)**	**91%**	**<0.001**
** *Other* **		
N6-Acetyllysine	−4%	0.86

## Data Availability

Data not contained within the article are available through the authors upon request.

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
