# Peer review of "High-Resolution Magic-Angle Spinning Nuclear Magnetic Resonance Identifies Impairment of Metabolism by T-2 Toxin, in Relation to Toxicity, in Zebrafish Embryo Model"

_toxins, 2024, doi:10.3390/toxins16100424_

Round 1

Reviewer 1 Report

Comments and Suggestions for Authors

The manuscript discusses the NMS-based exploration of T2 toxicological effects on zebrafish embryos. The work is well-structured, scientifically robust, and generally engaging for readers. The results are presented clearly and consistently discussed. The conclusion is strong and clear. The methods are adequately described. I did not identify any serious pitfalls, so I fully support its acceptance in its present form.

As a very minor suggestion, the quality and resolution of the figures could be improved. I guess this is an editing issue and could be addressed along the process.

Author Response

Comment: As a very minor suggestion, the quality and resolution of the figures could be improved. I guess this is an editing issue and could be addressed along the process.

Response: We truly appreciate the reviewer’s comments, and minor suggestions for revising the manuscript. We have endeavored to improve the figure quality and resolution, and will anticipate being requested to do so by the editors prior to final publication.

Reviewer 2 Report

Comments and Suggestions for Authors

Authors have described cellular, molecular, and biochemical pathways associated with T-2 toxicity employing High-Resolution Magic-Angle Spinning Nuclear Magnetic Resonance in the Zebrafish Embryo Model. The authors have covered many aspects of determining the mechanism of action but the manuscript needs to be revised to make a greater impact on the audience. 

1. Make a flow diagram of the overall methodology. Such as starting from pre and post-treatment measurements.

2. How did the author measure the ROS in different organs?

3. Did the author separate the organs for doing the ROS and other biochemical assays?

4. The author should make a stacked diagram of NMR spectra and show the qualitative variation for pre and post-treatment.

5. What are the most influential metabolites showing an insight of the mechanism of action? Write down in conclusion.  

6. How this model can be related to the toxicity assessment in human.

Author Response

Comment 1:  Make a flow diagram of the overall methodology. Such as starting from pre and post-treatment measurements.

Response:  This is a great suggestion. A diagram of the workflow (Figure 1) of the methodology has been added to the revised manuscript.

Comment 2: How did the author measure the ROS in different organs?

Response: To clarify, ROS in organ systems was not quantified strictly speaking, but rather qualitatively assessed (as apparent increase or decrease, or detected versus not detected) based on relative fluorescence observed.

Comment 3: Did the author separate the organs for doing the ROS and other biochemical assays?

Response: No, ROS was observed by confocal fluorescence microscopy in the intact embryo (as shown in Figure 3). And, as per the previous comment, ROS was semi-quantitatively assessed (i.e., as apparent increase or decrease, or detected versus not detected).

Comment 4: The author should make a stacked diagram of NMR spectra and show the qualitative variation for pre and post-treatment.

Response: We agree that this would a good addition, and have added a figure (Figure 2), showing side-by-side comparison the NMR spectra for Control and T-2 Treated embryos, to the revised manuscript.

Comment 5: What are the most influential metabolites showing an insight of the mechanism of action? Write down in conclusion.  

Response: As mentioned in the Discussion and Conclusion, perhaps the most notable pathways and targets identified through metabolic profiling are (1) succinate dehydrogenase (SDH), as both a component of the TCA cycle, and as Complex II of the electron transport chain of oxidative phosphorylation; and (2) suggested role of Phase I and II detoxification pathways including, in the latter case, possible glucuronidation. Indeed, both of these are pointed out, and emphasized in the Conclusions, however, as per the reviewer’s comments, we have elaborated in the revised Conclusions the key metabolites associated with both including, in the case of SDH, altered levels of succinate and fumarate, as well as decreased Phe as means of circumventing inhibition of SDH (Succ > Fum), and in the case of Phase II glucuronidation, the altered levels of glucose and related metabolites (i.e., G6P and G1P) as precursors for UDPG.

Comment 6: How this model can be related to the toxicity assessment in human.

Response: Indeed, the zebrafish embryo has been widely suggested to serve as a potential model for vertebrate toxicity – including human health. This point has been added to both the revised Introduction and Conclusions sections.

Reviewer 3 Report

Comments and Suggestions for Authors

The article submitted on Metabolism by T-2 Toxin, in Relation to Toxicity, in the Zebrafish Embryo Model by HRMAS NMR is well planned, elaborated and presented, and it also uses a very promising methodology for toxicity studies. This is evidenced by other recent studies carried out by the same authors on different toxicants.

The article provides little-known aspects of the toxicity of the T-2 toxin, although it is a pity that the HT-2 toxin has not been studied in parallel, since both are present in foods made from cereals such as bread and beer, among others.

However, some observation must be made:

1.- Eliminate the paragraphs between lines 6-10 of the abstract and 29-38 of the introduction. They are generalist paragraphs, whose information is well known, which do not add value to the article.

2.- The article (abstract and conclusions) talks about the identification of possible biomarkers of exposure to the toxic substance. As this statement is only a very distant hypothesis, the authors of this article can refer to it in the discussion but not in the abstract and conclusions.

3.- It would be more practical for other researchers to find this article if the keywords did not fully coincide with the words in the title.

Author Response

Comment 1: Eliminate the paragraphs between lines 6-10 of the abstract and 29-38 of the introduction. They are generalist paragraphs, whose information is well known, which do not add value to the article.

Response: We appreciate the reviewer’s comment that this information, regarding T-2 toxin including its occurrence and toxicity, is likely well known to those in the field of mycotoxins. However, we hesitate to remove this information entirely for sake of those outside of the field of mycotoxicology, or those perhaps not familiar with this particular toxin, or the trichothecenes more generally. That said, we have endeavored in the revised manuscript to make this point (which introduces T-2 toxin to the broader audience) more concise.

Comment 2: The article (abstract and conclusions) talks about the identification of possible biomarkers of exposure to the toxic substance. As this statement is only a very distant hypothesis, the authors of this article can refer to it in the discussion but not in the abstract and conclusions.

Response: We appreciate the reviewer’s comment, and have deemphasized this point in both the revised Abstract and Conclusions.

Comment 3: It would be more practical for other researchers to find this article if the keywords did not fully coincide with the words in the title.

Response: We have modified keywords to include some that are not included in the title.